# Facile Fabrication of Multifunctional Transparent Flame-Retarded Hydrogel for Fire-Resistant Glass with Excellent Transparency, Fire Resistance and Anti-Ageing Property

**DOI:** 10.3390/polym14132716

**Published:** 2022-07-02

**Authors:** Feiyue Wang, Mengtao Cai, Long Yan, Jiahao Liao

**Affiliations:** Institute of Disaster Prevention Science and Safety, School of Civil Engineering, Central South University, Changsha 410075, China; wfyhn@163.com (F.W.); 194812293@csu.edu.cn (M.C.); 194812294@csu.edu.cn (J.L.)

**Keywords:** multifunctional transparent flame-retarded hydrogels, fire-resistant glass, fire resistance, anti-ageing property, transparency

## Abstract

Acrylamide-methacrylic acid copolymer named P (AM-co-MAA) was synthesized via aqueous solution polymerization, and then mixed with crosslinker, flame retardants and initiators to prepare multifunctional transparent flame-retarded hydrogels with transparency, fire resistance and anti-ageing property. The results show that the application of multifunctional transparent flame-retarded hydrogel imparts high level of transparency and excellent fire resistance to the fire-resistant glass, and the light transmittance and fire resistance of the flame-retarded hydrogel increases with the increasing mass ratio of AM to MAA in P(AM-co-MAA). When the mass ratio of AM to MAA is 4:1, the obtained P(AM-co-MAA) imparts the lowest backside temperature of 130 °C at 3600 s and highest light transmittance of 86.1% to the transparent flame-retarded hydrogel. TG and DSC analysis show that the addition of P(AM-co-MAA) increases the thermal stability of the transparent flame-retarded hydrogels due to the formation of numerous hydrogen bonds via the complexation between amide and carboxyl groups. Accelerated ageing test indicates that the transparent flame-retarded hydrogel containing P(AM-co-MAA) exerts durable fire resistance and transparency, and the ageing resistance of the transparent flame-retarded hydrogel depends on the mass ratio of AM to MAA in P(AM-co-MAA). Therefore, this study provides a promising strategy to prepare a novel multifunctional transparent flame-retarded hydrogel with excellent light transmittance, fire resistance and anti-ageing properties.

## 1. Introduction

Superabsorbent hydrogels are cross-linked hydrophilic polymer networks with high swelling capacity in the aqueous medium and can absorb water up to hundreds of times greater than the polymer weight [1,2]. The high swelling capacity of superabsorbent hydrogels and the high specific heat capacity of water impart great advantages to superabsorbent hydrogels in fire resistance and extinguishing. Compared with traditional flame retardant materials, superabsorbent hydrogels can maintain lower temperature when exposed to flames due to the high water content, which offers the advantages of high heat insulation, cooling and water retention performance, and thus are widely used in fire prevention [3,4]. Apart from the fire extinguishing field, the superabsorbent hydrogels are widely used as fire-resistant interlayer to prepare perfusion fire-resistant glass with excellent heat insulation performance, which can greatly improve the fire safety of buildings via suppressing heat radiation, smoke and flame when exposed to flame or high temperature [5,6].

However, the fire resistance of the transparent flame-retarded hydrogel will degrade after the evaporation of water from the hydrogel. Therefore, it is crucial to enhance the fire resistance of transparent flame-retarded hydrogels via chemical modification of hydrogel network and physical adding flame retardants. Ammonium polyphosphate (APP), ammonium dihydrogen phosphate (ADP), ammonium hydrogen phosphate and other phosphorus-based flame retardants are widely used in the field of flame-retarded hydrogels to protect the hydrogel from burning [7]. Zheng et al. introduced basalt fibers and ammonium polyphosphate (APP) into polyacrylamide hydrogels to improve the fire resistance properties and found that APP can effectively delay the thermal decomposition process of the hydrogel by forming an expanded char layer [8]. Chen et al. introduced modified ammonium polyphosphate into sodium alginate/poly(acrylamide-stearyl methacrylate copolymer) hydrogels to improve the thermal stability and fire resistance of the hydrogels [9]. The introduction of phosphorus-based flame retardants in hydrogels is a promising way to enhance the fire resistance performance. Moreover, the traditional phosphorus-based flame retardants will reduce the hydrolytic stability of polymers due to the lack of metal cations [10], which is anticipated to impart super ageing resistance to the resulting flame-retarded hydrogels. To the best of our knowledge, few efforts focused on the study and application of the mixture of APP and ADP in transparent flame-retarded hydrogels.

Polyacrylamide (PAM) is widely used as polymer network for preparing transparent flame-retarded hydrogels. However, the transparent flame-retarded hydrogel obtained from polyacrylamide (PAM) exhibits insufficient hydrolytic stability and poor weather resistance [11]. The polymethylacrylic acid (PMAA) is used in various industries as stabilizer, thickener and flocculant in dispersion systems for the synthesis of water-soluble polymers [12,13,14]. Superabsorbent hydrogels based on methacrylic acid and acrylamide (P(AM-co-MAA)) are focused for the fact that the properties of transparent flame-retarded hydrogel can be optimized by varying the composition of P(AM-co-MAA) in the hydrogels [15]. However, few efforts have devoted to the fire resistance and anti-ageing properties of hydrogels based on P(AM-co-MAA). It is significant to study the effect of P(AM-co-MAA) on the fire resistance and ageing resistance of the hydrogels. Therefore, the co-polymerization of acrylic amide (AM) and methacrylic acid (MAA) is expected to reduce light scattering and produce transparent flame-retarded hydrogel [16].

In this paper, P(AM-co-MAA) was synthesized via aqueous solution polymerization, and then applied to fabricate a novel multifunctional transparent flame-retarded hydrogel for fire-resistant glass. The effect of P(AM-co-MAA) on the optical transparency, fire resistance, thermal stability and anti-ageing properties of the transparent flame-retarded hydrogels was carefully investigated, and the possible mechanism of flame-retardant for multifunctional transparent flame-retarded hydrogel is also proposed.

## 2. Experimental

### 2.1. Materials

Acrylamide (AM, purity: 99.0%), methacrylic acid (MAA, purity: 99.0%), ammonium persulfate (APS, purity: 99.99%), N, N-methylene bisacrylamide (MBA, purity: 99.5%) were supplied from Aladdin Chemistry Co., Ltd. (Shanghai, China). Sodium hydroxide (NaOH, purity: 98.0%) was obtained by Chengdu Cologne Chemical Co., Ltd. (Chengdu, China). Sodium metabisulfite (Na_2_S_2_O_5_, purity: 96.0%), ammonium dihydrogen phosphate ((NH_4_)H_2_PO_4_, purity: 99.0%), urea (H_2_NCONH_2_, purity: 99.0%), glycerol (C_2_H_8_O_3_, purity: 99.0%) were purchased from Tianjin Zhiyuan Chemical Reagent Co., Ltd. (Tianjin, China). Ammonium polyphosphate (APP, purity: 99.0%, *n* < 20) was obtained from McLin Biotech Co., Ltd. (Shanghai, China).

### 2.2. Preparation of Multifunctional Transparent Flame-Retarded Hydrogel

A series of transparent flame-retarded hydrogels with different composition were obtained by copolymerization of acrylic amide and methacrylic acid as shown in Table 1. First, a quantity of sodium hydroxide was dissolved in deionized water under ice bath conditions, and then methacrylic acid was slowly added to the sodium hydroxide solution with stirring to obtain sodium methacrylic acid solution of 75.0% neutralization, as shown in Figure 1. Then, a certain amount of acrylamide, *N,N*’-methylene bisacrylamide and flame retardant solution consisted of 5% ammonium polyphosphate, 4% ammonium dihydrogen phosphate, 1% urea and 3% glycerol were mixed and then stirred evenly. Finally, the mixed solution was subjected to copolymerization at 70 °C for 0.5 h to prepare acrylamide-methacrylic acid copolymer (P (AM-co-MAA)). The procedure for the synthesis of transparent flame-retarded hydrogel is shown in Figure 1. The theoretical mass content of phosphorus element in the hydrogel is 2.68%.

### 2.3. Preparation of Fire-Resistant Glass

The mixed transparent fire-resistant solution was kept for 24 h and then the mixed transparent fire-resistant solution was filtered, after which the sodium metabisulfite and ammonium persulfate were sequentially added to the mixed solution with constantly stirring. The fire-resistant glass consists of five layers of original glass sheets, each of which is 0.5 cm thick. The thickness of the polymer layer is 1.5 cm. After the addition of sodium metabisulfite and ammonium persulfate solution, the mixture was poured into the fire-resistant glass with a size of 15 cm × 15 cm × 2.5 cm by siphoning at room temperature, and then the transparent flame-retarded hydrogel sandwich was cured at 30–40 °C for 24–48 h to obtain the composite fire-resistant glass, which was marked as H-0~H-6.

### 2.4. Measurements and Characterization

#### 2.4.1. Fourier Transform Infrared Spectroscopy (FTIR) and ^1^H NMR Analysis

The transparent flame-retarded hydrogels were kept at 50 °C for 24 h in an oven, after which the obtained transparent flame-retarded hydrogels were characterized by an iCAN9 FTIR spectrometer (Tianjin Energy Spectrum Technology Co., Ltd., Tianjin, China). The samples were tested with KBr pellets in the wavenumber range of 4000–500 cm^−1^. ^1^H NMR experiment was taken on AVANCE400 Bruker with a proton frequency of 500.13 MHz at 298 K. D_2_O was used as solvent instead of water to weaken the water signal.

#### 2.4.2. Optical Transparency Analysis

The optical transparency was analyzed by LS183-108H transmittance meter (Shenzhen Lianhuicheng Technology Co., Ltd., Shenzhen, China). The light transmittance of the fire-resistant glass applying the transparent flame-retarded hydrogel was tested at several different locations of the sample. The average value of the light transmittance was obtained by averaging the light transmittance of multiple locations of the fire-resistant glass.

#### 2.4.3. Heat Insulation Test

The heat insulation performance of the fire-resistant glass applying multifunctional transparent flame-retarded hydrogel was tested using a cone heater at a heat flux of 50 kW/m^2^. The heat insulation test was carried out using the 6810F cone heater (Suzhou Yangyi wolch Testing Technology Co., Ltd., Suzhou, China), and the date of the backside temperature of the fire-resistant glass was recorded by temperature acquisition instrument and K-type thermocouple. 

#### 2.4.4. Thermo-Gravimetric Analysis

The tested transparent flame-retarded hydrogel samples were kept at 50 °C for 24 h in an oven. The TG analysis were carried out using TGA/SD-TA851 thermogravimetric analyzer (Mettretoli Instruments Co., Ltd., Zurich, Switzerland). The tested transparent flame-retarded hydrogel with a weight of 3–5 mg was tested at the heating rate of 10 °C/min from 25 °C to 600 °C under nitrogen atmosphere of 40 mL/min.

#### 2.4.5. Differential Scanning Calorimetry Analysis

Before the Differential scanning calorimetry (DSC) test, the tested transparent flame-retarded hydrogel samples were kept at 50 °C for 24 h in an oven. The differential scanning calorimetry analysis was performed by DSC823e (Mettler Toledo International Trading Co., Ltd., Shanghai, China). The samples were heated at a heating rate of 10 °C/min from 25 °C to 600 °C under a nitrogen atmosphere of 40 mL/min.

#### 2.4.6. Accelerated Ageing Test

The accelerated ageing test was carried out using the ASTM G154-2006 UV barometer (Dongguan Haoran Experimental Equipment Co., Ltd., Dongguan, China). The light transmittance of the samples was measured and recorded before and after UV exposure. The temperature of the sample was maintained at 45 °C ± 5 °C and the irradiation intensity of the UV lamp was 0.76 W/(m^2^·nm). The changes of the light transmittance and heat insulation performance of the transparent flame-retarded hydrogel after UV irradiation time at 24 h, 72 h, 120 h and 168 h were observed. 

## 3. Results and Discussion

### 3.1. Characterization of Poly (Acrylamide -Co-Methacrylic Acid)

As shown in Figure 2, the chemical structure of the polyacrylamide (PAM) and acrylamide -co-methacrylic acid copolymer (P (AM-co-MAA)) transparent flame-retarded hydrogels were characterized by FTIR analysis. It can be seen from Figure 2 that all samples show the similar characteristic peaks, including the mixed absorption stretching peak of N–H and O–H groups at 3410 cm^−1^ and the C=O group at 1672 cm^−1^ and C–N group at 1454 cm^−1^. For the spectrum of the transparent flame-retarded hydrogel based on P(AM-co-MAA), the infrared spectra of H-1, H-2 and H-3 samples containing P (AM-co-MAA) show two new characteristic peaks which are assigned to COO^–^ groups at 1542 cm^−1^ and 1560 cm^−1^, indicating the occurrence of copolymerization between acrylic amide and methacrylic acid as shown in Figure 1 [17,18,19].

To further confirm the chemical structure of P (AM-co-MAA), ^1^H NMR measurement was performed (D_2_O as the testing solvent) and the corresponding results are shown in Figure 3. The peak at δ 1.72 is typical resonance of the methylene (CH_2_) in ^1^H NMR spectrum of P (AM-co-MAA), further indicating the occurrence of copolymerization between AM and MAA.

### 3.2. Optical Transparency Analysis

The light transmittance and digital photos of the transparent flame-retarded hydrogels with different composition of poly (methacrylic acid-acrylamide) (P(AM-co-MAA)) are displayed in Figure 4. It can be seen that the clear observation of the logo under the fire-resistant glass indicates the indeed high transparency of the transparent flame-retarded hydrogel sandwich applied in the fire-resistant glass. Compared to the H-0 sample, the light transmittance of the transparent flame-retarded hydrogels containing P(AM-co-MAA) gradually decrease with increasing methacrylic acid content in P(AM-co-MAA). Especially, when the percentage of methacrylic acid in P(AM-co-MAA) exceeds 50%, the light transmittance of H-5 sample drops to 82.6%, which is ascribed to the hydrophobic interactions between methyl groups and the entanglement of copolymer chains, leading to an increase of the refractive index of the hydrogel interlayer of fire-resistant glass. The decrease of the light transmittance as the content of methacrylic acid increases results from the occurrence of a microphase separation process in the hydrogel, which is related to the aggregation of MAA units [20]. Moreover, the electrostatic repulsion of the sub-chain with the increase of the ionic carboxyl fraction will decrease compatibility between the hydrogel networks with the ionized flame retardants, resulting in the light scattering as well as the decrease of transparency.

### 3.3. Fire Resistance Analysis

The cone heater was used to test the heat insulation performance of the fire-resistant glass applying the transparent flame-retarded hydrogel. The backside temperature curves of the samples obtained from the heat insulation test are shown in Figure 5. As shown in Figure 5, it is evident that the backside temperature rising process can be divided into three stages, including the initial temperature rise stage, temperature stabilization stage and temperature surge stage. The backside temperature of the H-0 sample rapidly exceeds 180 °C after heat insulation test of 3600 s, which is due to the low compatibility between acrylamide-based fire-resistant hydrogel and the flame retardants, thus reducing the fire resistance of transparent fire-resistant hydrogel. However, it can be seen that there is a decrease of the backside temperature rising speed of H-1 to H-5 samples and the samples containing P(AM-co-MAA) have a lower final backside temperature compared to polyacrylamide hydrogel, indicating that the samples containing P(AM-co-MAA) show better heat insulation property compared to the H-0 sample and the introduction of methacrylic acid is beneficial to enhance the fire resistance of the transparent flame-retarded hydrogels. The fire resistance of the transparent flame-retarded hydrogels depends on the composition of P(AM-co-MAA). The fire resistance of the samples increases with the increasing mass ratio of acrylic amide to methacrylic acid in P(AM-co-MAA), which is ascribed to the enhanced hydrophobic interaction between the methyl groups resulting in a tighter polymer network that reduces the water retention properties of the transparent flame-retarded hydrogel [21]. Especially, the P(AM-co-MAA) with a mass ratio of 4:1 of acrylic amide to methacrylic acid imparts the lowest backside temperature of 130 °C at 3600 s to the H-1 sample, corresponding to the best heat insulation property among all samples, indicating that the introduction of lower content of MAA chain in P(AM-co-AA) is more beneficial for enhancing the fire resistance of the transparent flame-retarded hydrogel. 

As seen from Figure 6, the digital photos of the char residues from the heat insulation test indicates that the introduction of P(AM-co-MAA) is beneficial to enhance the intumescence and compactness of char residue, thus effectively enhancing the fire resistance of transparent flame-retarded hydrogels. Moreover, the char layer of the transparent flame-retarded hydrogel become more compact and continuous during the combustion process with the increasing content of MAA chain in P(AM-co-MAA). Especially, the H-1 sample shows the tallest intumescent char layer with a height of 4.5 cm, corresponding to the best heat insulation performance. Moreover, the decrease of expansion and foaming effect of the transparent flame-retarded hydrogel with the increasing of methacrylic acid unit in the copolymer is because that the increase of methacrylic acid unit content in the copolymer leads to an increase of hydrophobicity of the system, which reduces the interaction between the hydrogel polymer network and water, resulting in a decrease in fire resistance of the transparent fire-resistant hydrogel.

The possible flame-retardant mechanism of the transparent flame-retarded hydrogels containing P(AM-co-MAA) is shown in Figure 7. The backside temperature changes of the samples reflect the thermal phenomenon in the transparent flame-retarded hydrogel. During combustion, the free water in the transparent flame-retarded hydrogel rapidly evaporates and absorbs large amounts of heat to cool the temperature of the combustion zone. The flame retardants such as ammonium dihydrogen phosphate and ammonium polyphosphate in the hydrogel are decomposed to the esterified charcoal sources and polyphosphoric acid that can be used as a dehydrating agent to catalyze the crosslinking reaction of carbon-containing compounds with a formation of a molten char layer, including the reaction of inorganic acid with polyhydroxy compounds such as glycerol to form a carbonized layer with porous structure. Moreover, ammonium polyphosphate breaks and decomposes to produce cyclic phosphate esters at about 200 °C, and produces an unsaturated carbon rich structure at a high temperature. At the same time, the thermal degradation of amide groups and the carboxyl groups on the polymer chain and the flame retardants accompanied by the release of non-flammable gases such as ammonia and water vapor lead to the expansion process of the char layer. The formation of strong char layer after water evaporation provides an excellent heat insulation barrier to oxygen and combustible gases, and can effectively prevent the heat transfer.

### 3.4. Thermal Stability Analysis

The thermal behavior of the transparent flame-retarded hydrogels was analyzed by TG tester under a nitrogen atmosphere of 50 mL/min. The TG and DTG curves of the samples are presented in Figure 8, and the related thermal data is shown in Table 2. It can be seen from Figure 8 and Table 2 that there are three decomposition stages of H−0 sample in the temperature ranges of 25–210 °C, 210–350 °C and 350–600 °C. The first stage at 25–210 °C is dominated by the evaporation of polymer network bound water and evaporation of small molecules in polymer network. The polymer side-chains start to decompose at 210–350 °C, while the adjacent amide groups decompose into amide groups, ammonia and water with slight loss of mass. When the test temperature rises up to 350 °C, the polyacrylamide (PAM) backbone depolymerizes to imide, nitrile and CO_2_ concomitant with the formation of molten char layer. The residual weight of the H-0 sample reaches 23.16% at 600 °C.

Compared to the H-0 sample, the introduction of P(AM-co-MAA) greatly increases the char residual weight of the transparent flame-retarded hydrogels, which is ascribed to that more hydrogen bonds are formed via the complexation between acrylic amide (AM) and methacrylic acid (MAA) [22]. Meanwhile, a large number of hydroxyl groups are introduced into the copolymer, which helps to form more hydrogen bonds and prevent the dissociation of acidic groups on the polymer chains. The char residual weight of the transparent flame-retarded hydrogels based on P(AM-co-MAA) gradually decreases with increasing methacrylic acid content in P(AM-co-MAA), which is attributed to the decrease of water retention property arising from the enhancement of ionization of the carboxyl groups and electrostatic repulsion, resulting in the network structure of hydrogel swells and releases the internal water [23,24]. Especially, the H-1 sample exhibits the highest char residue of 36.4% at 600 °C among all the samples, which may be ascribed to the hydrophobic interaction of α-methyl groups in methacrylic acid enables easier formation and greater stability of intrachain hydrogen bonds [25,26]. In summary, the introduction of P (AM-co-MAA) produced excellent thermal stability and carbonization effect of the transparent flame-retarded hydrogel, which is consistent with the heat insulation performance.

The DSC curves of the samples are shown in Figure 9. As can be seen from Figure 9 that the DSC curves of the transparent flame-retarded hydrogels show a clear endothermic and an exothermic process. For H-0 hydrogel containing polyacrylamide (PAM), there are four endothermic peaks at 100–450 °C. The endothermic peaks at 120–210 °C is attributed to the decomposition of amide groups and flame retardants. The endothermic peak at 220–300 °C is ascribed to the dehydration condensation of adjacent amide groups. As the temperature rises to 300 °C, the main chain of polyacrylamide (PAM) is depolymerized into imide, nitrile and CO_2_. The exothermic peak at 400–500 °C may be due to the release of molecular water during the polymerization and recrystallization processes of hydrogel decomposition into CO_2_ and long-chain polymers. Moreover, the exothermic peak after 500 °C may be attributed to the exothermic oxidation of the transparent flame-retarded hydrogel and the decomposition of the polymer backbone. In particular, the DSC curve of the H-0 sample appears two major endothermic peaks at 160 °C and 248 °C, respectively, while the corresponding endothermic peaks of the H-1 sample are shifted to 186 °C and 320 °C, respectively. The reaction between adjacent acrylic amide and methacrylic acid units on the main chain of the transparent flame-retarded hydrogel based on P(AM-co-MAA) in the H-1 sample forms a six-membered imide ring and cross-linked structure compared to the H-0 sample [27], thus showing higher temperature of endothermic peak and better thermal stability. The exothermic process after 450 °C is mainly attributed to the char formation of the hydrogel, and the sample of H-1 exhibits a stronger exothermic process than the H-0 sample at 300–600 °C due to its better char formation ability, as supported by TG analysis.

### 3.5. Ageing Resistance Analysis

Figure 10 shows the digital photographs and light transmittance of the samples at various accelerated ageing times. As can be seen from Figure 10 that the H-0 sample shows an obvious decrease in light transmittance along with the appearance of bubbles and turbidity after the accelerated ageing test, which come from the voids generated by the volume contraction of hydrogel under UV irradiation. The loss of transparency of the transparent flame-retarded hydrogel is attributed to the decomposition of the transparent flame-retarded hydrogel and the precipitation of flame retardants under UV irradiation. Specifically, the turbidity of H-0 hydrogel interlayer is ascribed to the intramolecular and intermolecular imidization of polyacrylamide yield partially or completely insoluble products [28].

Compared to the H-0 sample, the decline degree of optical transparency of the P (AM-co-MAA) hydrogels gradually decreases with the decreasing mass ratio of acrylic amide to methacrylic acid in P (AM-co-MAA) under the same ageing treatment. However, the H-5 and H-6 samples with high content of methacrylic acid show discoloration after 168 h accelerated ageing treatment, which is mainly caused by the chromophores produced by UV radiation [29], indicating that P(AM-co-MAA) flame-retarded hydrogels with a high content of methacrylic acid have a poor durable light transmittance.

Figure 11 shows the degradation degree of light transmittance for the samples after 168 h accelerated ageing time. The light transmittance of the H-0 sample decreases significantly up to 21.7% after the accelerated ageing time of 168 h. In contrast, when the mass ratio of acrylic amide to methacrylic acid
is 4:1 in P (AM-co-MAA), the light transmittance of the H-1 sample decreases by only 2.5%, indicating that the existence of P (AM-co-MAA) with low methacrylic acid (MAA) chain content can effectively improve the anti-ageing performance of the transparent flame-retarded hydrogel.

Figure 12 shows the backside temperature curves from the heat insulation test of the samples before and after accelerated ageing test, and the date of the backside temperature variation of the unaged samples and aged samples are shown in Table 3. It can be seen from Figure 12 that the fire resistance of the H-0 sample is significantly degraded after the accelerated ageing treatment. As shown in Table 3, the degradation degree of the backside temperature at 3600 s before and after accelerated ageing test increases with decreasing mass ratio of acrylic amide to methacrylic acid in P (AM-co-MAA), which indicates that the introduction of P(AM-co-MAA) with low content of methacrylic acid can more effectively improve the durable fire resistance and anti-aging properties of the transparent flame-retarded hydrogels. On the one hand, the improvement of the anti-aging performance is ascribed to the stronger hydrolytic stability of P (AM-co-AA). On the other hand, it is because that the addition of methacrylic acid enhances the intermolecular linkage. Particularly, the P(AM-co-MAA) with a mass ratio of 4: 1 of acrylic amide to methacrylic acid imparts the best anti-ageing performance to the transparent flame-retarded hydrogel.

To further explore the anti-aging mechanism of the transparent flame-retarded hydrogel, the unaged and aged H-0 and H-1 hydrogels with different ageing times of 0 h, 24 h, 72 h and 168 h were examined by FTIR analysis. The FTIR spectra of the H-0 and H-1 samples treated with different ageing times are shown in Figure 13. The intensity variation of the characteristic peaks of the main groups for transparent flame-retarded hydrogels under ageing treatment are shown in Table 4. As can be seen in Figure 13, similar characteristic peaks are observed in the spectra of the samples before and after accelerated ageing test. The intensities of some characteristic peaks show obvious differences. As seen from Figure 13a that the intensities of C=O group at 1672 cm^−1^ and C–O group at 1150 cm^−1^ are gradually strengthened with increasing ageing times due to the hydrolysis of amide groups upon exposure to UV radiation, resulting in the separation of water and polymer network as well as the decrease in light transmittance of the transparent flame-retarded hydrogel. The peak of P=O groups at 1307 cm^−1^ gradually strengthens with increasing ageing times owing to the degradation of flame retardants in the polymer network [30]. As can be seen from Figure 13b, compared to the H-0 sample, the intensity of C=O group in the spectrum of the H-1 sample has almost no change before and after ageing test, which is ascribed to inactive amide groups enclosed by adjacent carboxylate groups and the high hydrolytic stability of carboxylates and carboxylic acids [28]. In summary, the introduction of P(AM-co-MAA) slows down the hydrolysis reaction of the amide group and thus enhances the ageing resistance, which is consistent with the durable fire resistance and light transmittance.

## 4. Conclusions

In this paper, poly (acrylamide -co-methacrylic acid) (P(AM-co-AA)) was successfully synthesized using acrylic amide (AM) and methacrylic acid (MAA) as monomers, *N,N*’-methylenebisacrylamide (MBA) as crosslinker, ammonium persulfate and sodium metabisulfite as initiators via the solution polymerization and were comprehensively characterized by FTIR, optical transparency analysis, TG and DSC analysis and anti-aging analysis. Then, P(AM-co-AA) was mixed with flame retardants including ammonium polyphosphate, ammonium dihydrogen phosphate, urea and glycerol to fabricate multifunctional transparent flame-retarded hydrogels. The effect of the mass ratio of acrylamide (AM) to methacrylic acid (MAA) in P(AM-co-MAA) transparent fire-resistant hydrogel on optical transparency, thermal stability, fire resistance and ageing properties was evaluated by various analytical methods. The result of optical transparency analysis shows that the light transmittance of the transparent flam-retarded hydrogel gradually decreases with the increase of polymethyl (PMAA) fragments in P (AM-co-MAA). In particular, when the mass ratio of acrylic amide to methacrylic acid is 4:1, the obtained P (AM-co-AA) imparts the H-1 sample the highest transmittance of 86.1%. The result of the fire resistance analysis shows that the introduction of methacrylic acid significantly enhances the thermal stability and char formation ability of the transparent flam-retarded hydrogel, but the synergistic fire resistance effect between the transparent flam-retarded hydrogel and compound flame retardants decreases continuously with the increase of the methacrylic acid content, and the sample of H-1 shows the best fire resistance. The backside temperature of the H-1 sample is reduced by 60.8% at 3600 s compared to the H-0 sample. Thermal analysis shows that the heat insulation properties and char formation ability of the hydrogels were significantly enhanced due to the complexation of acrylamide and methacrylic acid (MAA) monomer units as well as the hydrophobic interaction of the methyl groups on the polymer chains, resulting in easier formation and better stabilization of the intramolecular hydrogen bonds. Accelerated aging test shows that after 168 h accelerated aging test, the light transmittance and backside temperature change at 3600 s of H-1 after aging is 16.8% and 84 °C lower than H-0, respectively, which shows the most excellent durable fire resistance and aging resistance performance. Moreover, the enhanced effect of P(AM-co-MAA) in the hydrogel depends on the mass ratio of acrylic amide to methacrylic acid in P(AM-co-MAA), and the P(AM-co-AA) with a 4:1 mass ratio of acrylic amide to methacrylic acid shows the superior fire resistance, thermal stability and anti-ageing property. In general, P(AM-co-MAA) can be applied to fabricate a novel multifunctional transparent flame-retarded hydrogel with excellent transparency, fire resistance, thermal stability and ageing resistance.

## Data Availability

The data presented in this study are available upon request from the corresponding author.

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
