# Peer review of "Facile Fabrication of Multifunctional Transparent Flame-Retarded Hydrogel for Fire-Resistant Glass with Excellent Transparency, Fire Resistance and Anti-Ageing Property"

_polymers, 2022, doi:10.3390/polym14132716_

Round 1
Reviewer 1 Report
This work is potentially exciting and although the article is interesting, there are problems with the authors' style, which, in places, is a little tedious and not engaging for the reader.
Please describe Figure 1 adequately in the caption and define what a, b, c and d are in fig 1.
Can you please add a copolymerization scheme separate from figure 1?
Please redraw the crosslinked network of the P (AM-co-MAA) structure; the bonds look distorted.
You mentioned the Flame retardants were added to 13%, which is too much for me. Can you calculate the theoretical and practical P% concentration after adding flame retardants?
How you purify your acrylamide-methacrylic acid copolymer (AM-co-MAA)). Do you have any NMR data to prove your copolymer rection?
Please specify whether Thermo-gravimetric analysis measurements were performed under a nitrogen or air atmosphere?
Please summarize your TGA and DSC result in the table for ease of the reader?
Can you measure the Limiting Oxygen Index (LOI) to support your flame retardancy? Direct insertion probe mass spectrometry (DIP-MS), Microscale Combustion Calorimeter (MCC), and Cone calorimetry to support your mechanism.
Can you get Rheological measurements of your gel?
Reviewer 2 Report
The paper is very practical in the content and suitable for publication after the authors revised the manuscript.
1. On line 89, The formula of ammonium dihydrogen phosphate is wrong. The authors should correct it.
2. On line 89, what are urea and glycorol used for? The authors should describe it.
3. In Table 1, what is the composition of the flame retardant? What is the proportion of each composition?
4. On line 112, the authors should describe the structure of the fire-resistant glass sandwich, for example, the thickness of the polymer layer.
5. On line 245, the authors should describe the role of charring agent in flame retardant mechanism.
6. On line 317, the authors describe the loss of transparency of the hydrogel is attributed. The effect is not so clear. The authors should provide more explanations.
